# Circuit as Set of Points

**Jialv Zou**[†1], **Xinggang Wang**[‡1], **Jiahao Guo**[1], **Wenyu Liu**[1], **Qian Zhang**[2], **Chang Huang**[2]

[1]School of EIC, Huazhong University of Science and Technology
[2]Horizon Robotics

## Abstract

As the size of circuit designs continues to grow rapidly, artificial intelligence technologies are being extensively used in Electronic Design Automation (EDA) to assist with circuit design. Placement and routing are the most time-consuming parts of the physical design process, and how to quickly evaluate the placement has become a hot research topic. Prior works either transformed circuit designs into images using hand-crafted methods and then used Convolutional Neural Networks (CNN) to extract features, which are limited by the quality of the hand-crafted methods and could not achieve end-to-end training, or treated the circuit design as a graph structure and used Graph Neural Networks (GNN) to extract features, which require time-consuming preprocessing. In our work, we propose a novel perspective for circuit design by treating circuit components as point clouds and using Transformer-based point cloud perception methods to extract features from the circuit. This approach enables direct feature extraction from raw data without any preprocessing, allows for end-to-end training, and results in high performance. Experimental results show that our method achieves state-of-the-art performance in congestion prediction tasks on both the CircuitNet and ISPD2015 datasets, as well as in design rule check (DRC) violation prediction tasks on the CircuitNet dataset. Our method establishes a bridge between the relatively mature point cloud perception methods and the fast-developing EDA algorithms, enabling us to leverage more collective intelligence to solve this task. To facilitate the research of open EDA design, source codes and pre-trained models are released at `https://github.com/hustvl/circuitformer`.

## 1 Introduction

In recent years, the demand for semiconductor chips for various applications has been increasing, and the need to accelerate the chip design and manufacturing process has become increasingly urgent. And with the development of semiconductor technology, the scale of very-large-scale integrated (VLSI) circuits is growing exponentially, challenging the scalability and reliability of the IC design process. As a result, the industry urgently needs more efficient Electronic Design Automation (EDA) algorithms and tools to enable the processing of huge exploration spaces in a shorter period of time. The layout and routing phase are particularly important in the overall EDA flow. As feature size shrinks, routability becomes a more important constraint on the manufacturability of VLSI designs. Routing congestion can significantly affect PPA (Performance, Power, Area) metrics, and design rule violations can directly lead to unmanufactured designs, but there is no way to obtain an accurate congestion and design rule violations map without conducting placement and routing attempts, which can lead to longer design cycles and possibly some unpleasant surprises. Therefore, accurate and rapid evaluation of placement in advance of routing will be very important, and the exploration of routability becomes one of the key issues in modern placement for VLSI circuits [27].

---

[†] This work was done when Jialv Zou was interning at Horizon Robotics. [‡] Xinggang Wang (`xgwang@hust.edu.cn`) is the corresponding author.

37th Conference on Neural Information Processing Systems (NeurIPS 2023).

Before global placement, the circuit design is represented as a netlist consisting of nodes and nets, where nodes represent specific components and nets represent link relationships between nodes, containing topological information about the design. After the global placement, we will get information about the specific location of each node, containing the geometric information. How to use this information to make a fast and accurate evaluation of the placement is currently divided into roughly two approaches: (1) using hand-crafted features to rasterize the circuit and thus convert them into images. Then, using CNN [24, 38] or Generative Adversarial Nets (GAN) [42, 1, 45] to perform feature extraction on the circuit. (2) converting the circuit design into a graph structure and using Graph Neural Networks (GNN) [20, 35, 15] to perform feature extraction on the circuit design.

However, both methods have their advantages and disadvantages. The **CNN-based methods** use some hand-crafted features (e.g., RUDY [33], node density, pin density) to rasterize the circuit and perform feature extraction on it, where the grids are treated as pixels. Then, use classical image segmentation networks in computer vision (e.g., UNet [32] and FCN [25]) to do the pixel-wise prediction of labels (e.g., congestion and DRC violation). This method is simple and efficient, however, the effectiveness of the hand-crafted features limits the network's performance, and it can not achieve end-to-end training. The **GNN-based methods** involves converting the circuit design into a graph with vertices and edges, where nodes in the circuit are represented as vertices and nets are represented as edges. GNN networks are then used to extract features from the graph, making it suitable for downstream tasks. Although they have small parameters and fast inference speeds, it is worth noting that even though the time consumption of constructing the graph structure has been proven to be linearly related to the design scale, the design scale of chips is very large. The number of nodes is typically in the hundreds of thousands, while the number of edges can be in the millions, so the process of constructing the graph structure can often be very time-consuming. In addition, these methods also use hand-crafted features when processing geometric information, which also suffers from the aforementioned problems.

To address the above issues, in this paper, we approach the task of feature extraction in circuit design from a novel perspective by treating circuit components as point clouds and using a Transformer to solve the task, i.e., CircuitFormer. CircuitFormer avoids the two core pain points of the above two methods: (1) there is no need to use hand-crafted features, and end-to-end training can be achieved; (2) there is no need for any preprocessing, which greatly reduces the time of the entire process.

As shown in Fig. 1, our model is divided into two parts: an **encoder** and a **decoder**. The encoder extracts features from the placement information of the circuit design, while the decoder adapts to downstream tasks using a conventional backbone and segmentation head, similar to the CNN-based methods.

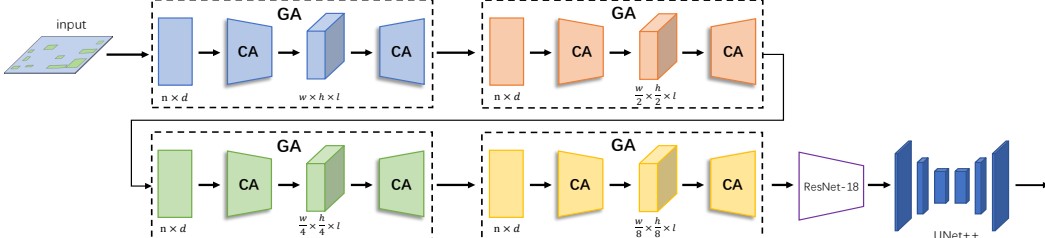

Figure 1: The architecture of CircuitFormer: The left half of the model is the encoder, which is composed of four grid-based attention (GA) modules with different grid scales. Each GA module extracts circuit features by performing cross-attention (CA) twice with the latent codes, using ConvFFN for information exchange across grids in the middle. For more details on the GA module, please refer to Fig. 2. The right half of the model is the decoder, which is used for pixel-wise prediction.

In **encoder**, in order to model the relationship between nodes, to compensate to some extent the disadvantage that geometric methods cannot use topological information, and to enable the exchange of information across the grids, we introduce the attention mechanism. Considering that most perception tasks for point clouds are focused on 3D point clouds, but circuits can be abstracted as 2D point clouds without a Z-axis, we propose an improvement to the voxel-based set attention (VSA)

module [16], adapting it for use in 2D scenarios to extract 2D node features. In Sec. 3.1 we will provide a detailed introduction.

The **decoder** consists of a backbone and a segmentation head. We use ResNet-18 [17] as the backbone to further refine the rasterized features extracted by the encoder. Finally, we attach a UNet++ [46] as the segmentation head to make pixel-wise predictions.

Our experiments were based on the publicly available CircuitNet [5] dataset and ISPD2015 [4] dataset. Whether it is the congestion prediction task or the DRC violation prediction task, the pixel values of their labels are continuous values between 0 and 1. The label distribution is highly imbalanced, exhibiting a skewed distribution with a long tail. The imbalance of label distribution can make it difficult for a model to handle corner cases, leading to overfitting and reducing the accuracy and robustness of the model. For the regression on imbalanced datasets, we adopted label distribution smoothing (LDS) [39] to re-weight the loss function and improve the model's performance.

In summary, our contributions are:

- Our work takes a novel perspective on circuit design by treating circuit components as point clouds and using a point cloud perception network to extract features for circuit design. To the best of our knowledge, our work is the first to abstract the task of circuit design feature extraction as a point cloud perception task. This bridges the gap between the two fields, allowing us to transfer certain structures and conclusions from mature point cloud perception methods to the developing EDA domain.

- Our method surpasses hand-crafted features in terms of feature extraction and enables end-to-end training, making it superior to CNN-based methods. Additionally, unlike GNN-based methods, our approach eliminates the need for preprocessing, significantly reducing processing time.

- Experiments validated the superior performance of our method on the CircuitNet and ISPD2015 datasets. In the congestion prediction task, our method surpassed the baseline [24] by an average of $3.8\%$ and $1.5\%$ on the CircuitNet dataset and ISPD2015 dataset respectively, based on the average of Pearson, Spearman, and Kendall correlation coefficients, achieving a new SOTA. For the DRC violation prediction task, we also achieved the best performance in most of the metrics. Our work has the potential to serve as a universal feature extractor for chip design, which can adapt to a wider range of downstream tasks by changing the decoder.

## 2 Related Work

### 2.1 CNN-based Methods

The core idea of CNN-based methods is to partition the circuit design into grids and extract features of the components in each grid using some hand-crafted methods. These methods treat the circuit as an image and the grids as pixels. Then, [24, 38] use features as the input to the segmentation network and performs pixel-wise prediction. [42, 1, 45] transform the task into generating a congestion map from features and uses a Conditional Generative Adversarial Network (CGAN) [28] to solve it.

### 2.2 GNN-based Methods

The GNN-based methods primarily focus on the topology of circuit design, which involves reconstructing the circuit design as a graph with vertices and edges. In [21], nodes serve as vertices, while nets serve as edges, then GAT [35] is used to extract features. [9] uses GraphSAGE [15] to extract features from a graph with G-cells as nodes. These methods can only utilize the topological information of the circuit design and are unable to handle geometric information. Recent advancements in graph networks have expanded their capabilities to incorporate geometric information in addition to topological information. LHNN [36] considers the grid as internal nodes and the net as external nodes, constructing a homogeneous graph. CircuitGNN [40] generates a heterogeneous graph, treating both nodes and nets as vertices, with information being propagated along both topological and geometric edges.

## 2.3 Point Cloud Perception

Point cloud perception methods can be classified into two categories: point-based and voxel-based. Point-based methods [30, 31] follow the paradigm of sample, group, and fusion. They perform farthest point sampling in the point cloud, then group points within a certain range of each sampled point, and finally perform the fusion. These methods can preserve the structure of the point cloud without losing information. However, due to the irregularity of point clouds, such methods are often difficult to train in parallel and require a large amount of memory consumption. Voxel-based methods [44] assign each point to a specific voxel and extract local features from groups of points within each voxel. This significantly improves computational efficiency but can result in information loss. Transformers can model the relationships between points and have vast potential in point cloud perception tasks. However, due to the quadratic computational complexity of transformers, vanilla transformers cannot be directly applied to point clouds. Based on this, [43] performs local attention on a set of points after sampling to extract local point cloud relationships. [16] applies cross attention twice between the latent code and point cloud to compress the features into the latent space, reducing computational complexity.

## 3 Method

We focus on the congestion prediction task and design rule check (DRC) violation prediction task. Congestion is defined as the overflow of routing demand over available routing resources in the routing stage of the back-end design, and DRC violation, as the name suggests, refers to the location where design rules are violated. Both tasks can be abstracted as making grid-level predictions of the circuit based on the point-wise information (geometric and topological) of the circuit design. Specifically, we consider circuit components such as buffers, inverters, registers, and intellectual property (IP) cores as point clouds. Here, we do not differentiate between component categories but only focus on their fundamental geometric features, i.e., the center coordinates, width, and height of each circuit component.

### 3.1 Encoder

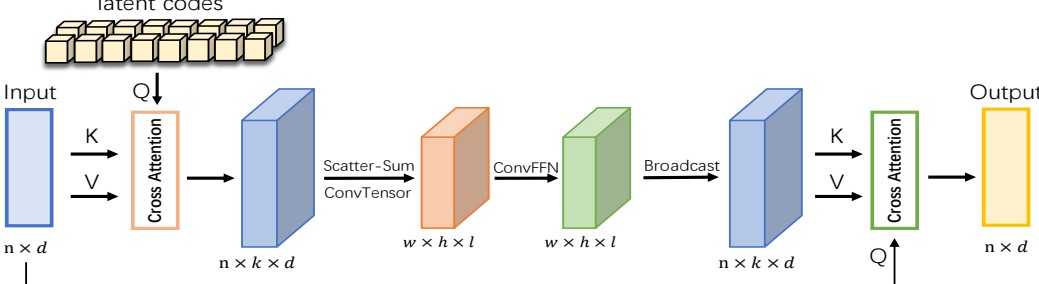

Figure 2: The architecture of grid-based attention (GA) module

**Multi-scale Grid-based Attention.** Due to the complexity of circuit designs, two nodes connected by a topological structure may be far apart in geometric positions. The hand-crafted method that only considers the geometric features of each node within a grid cannot handle the relationships between different grids. Therefore, we hope to introduce the attention mechanism to expand the receptive field, model long-range dependencies, and enable interaction between different grids. However, vanilla self-attention computation is quadratic, with a computational complexity of $\mathcal{O}(n^2 d)$, where $n$ represents the number of nodes, which is usually in the hundreds of thousands. Such high computational complexity is unacceptable. Inspired by [19, 18, 23, 16], we use a fixed-length, learnable latent code to perform two cross-attentions with the input features instead of one self-attention. This compresses the node information into a more compact latent space and significantly reduces the computational complexity. The validity of this method comes from the observation that the attention matrix is usually low-rank [11, 2].

The grid-based attention (GA) module is an improvement over the voxel-based set attention (VSA) module [16], adapted for use with the circuit, which is referred to as pseudo-2D point clouds in our work. The structure is shown in Fig. 2. Given an input set $P \in R^{n \times 4}$ where $n$ represents the number of nodes. The features of each node can be represented as $\{P_i = (x_i, y_i, w_i, h_i) : i = 1, ...n\}$, $x_i$ and $y_i$ represent the original coordinates, while $w_i$ and $h_i$ represent the width and height of the component, respectively. And we also have information on the position of corresponding points $C \in R^{n \times 2}$, $C = \{C_i = (\widetilde{x}_i, \widetilde{y}_i) : i = 1, ...n\}$, where $\widetilde{x}_i$ and $\widetilde{y}_i$ represent the normalized coordinates of the nodes. We encode the node features as keys $K \in R^{n \times d}$ and values $V \in R^{n \times d}$ through linear projection, same as vanilla Transformer. Randomly initialized learnable latent codes serve as queries $Q \in R^{k \times d}$ ($k \ll n$) and are fused with linearly projected node features to encode them into the latent space using cross-attention. After that, we obtain the point-wise feature $H \in R^{n \times k \times d}$.

Then, we convert the point-wise features to grid-wise features, the grid coordinates corresponding to each point can be represented as $G = \{ G_i = ([\frac{\widetilde{x}_i}{d_x}], [\frac{\widetilde{y}_i}{d_y}]) : i = 1, ...n) \}$, where $d_x$, $d_y$ is the grid size of two dimensions, and $[\cdot]$ is the floor function. By changing $d_x$, $d_y$, we can obtain raster features of different scales, so that we can perform multi-scale grid information exchange. To be more precise, we established four distinct grid scales, for $d_x = d_y = [1, 2, 4, 8]$. Next, we use the scatter-sum function to add point-wise features belonging to the same grid and obtain the grid-wise feature $A \in R^{m \times k \times d}$, where $m$ represents the number of non-empty grids. The scatter function is a CUDA kernel library that performs symmetric reduction on various matrix segments. The entire process is highly parallelized.

To introduce local inductive bias, based on the coordinates of the grids, we construct a sparse tensor $I \in R^{w \times h \times l}$ by assembling the grid-wise features into a image-like structure, where $w$ and $h$ respectively represent the number of grids in the two dimensions, and $l = k \times d$. To be specific, we pad the empty grids with zeros. After that, the feature $I$ is refined using a convolutional feed-forward network (ConvFFN) while enabling information exchange between grids, we use two layers of depth-wise convolution (DWConv) to achieve it. Datasets in circuit design are typically small in scale, and this approach of combining attention mechanisms with CNNs has been shown to be effective on small datasets [29]. This is because ConvFFN can introduce inductive bias to the model, which plays an important role in improving the convergence of transformer on small datasets, which are difficult to optimize.

To maintain consistency with the input dimensions, we broadcast the sparse tensor $I$ to generate the point-wise feature $\hat{H} \in R^{n \times k \times d}$ based on the grid indices. Finally, the refined point-wise feature $\hat{H}$ is used as the key-value pair, with the linear projection of the input serving as the query for cross-attention. This results in an output $O \in R^{n \times d}$ with the same dimensions as the input.

The encoder consists of four grid-based attention modules with different grid scales, which can aggregate multi-scale and global geometric information in a highly parallelized manner. It has a computational complexity of $\mathcal{O}(nkd)$, where $k \ll n$. At the end of the encoder, we use the scatter-sum operator to generate grid-wise features and combine them into an image $Y \in R^{W \times H \times D}$ according to the position coordinates of each grid, preparing it for further pixel-wise prediction. Here, $Y$ has the same width and height as the label.

## 3.2   Decoder

We did not put much emphasis on the design of the decoder. We utilized a ResNet-18 [17] as the backbone to further refine and downsample the features extracted by the encoder, and then employed a classic segmentation head, UNet++ [46], to achieve pixel-wise prediction.

## 3.3   Loss Function with Label Distribution Smoothing

Our label smoothing strategy follows the idea of re-weighting in imbalanced classification problems [3]. However, in the case where the labels are continuous values, the empirical label distribution may not reflect the true label distribution. This is because, unlike classification tasks, there are no hard boundaries between adjacent label values. When presented with limited samples for certain labels, the model prefers to learn features of labels with similar values.

Inspired by [39], we count the empirical density distribution of the labels in the training set, and then, convolve with it using a symmetric Gaussian kernel function to obtain the smoothed effective label

density distribution. The inverse of the square root of the smoothed distribution is used as the weight to do the re-weight of the loss function. The label distribution smooth calculation is shown below:

$$\tilde{p}(y') = \int_y k(y, y')p(y)dy \tag{1}$$

where $p(y)$ is the empirical density function of the label, we discretize the labels with a precision of 0.001 and count the frequency of the value $y$ to obtain $p(y)$. $\tilde{p}(y')$ is the effective density function of the label $y'$, and the weight corresponding to a label with value $y'$ is $\frac{1}{\sqrt{\tilde{p}(y')}}$. $k(y, y')$ is symmetric Gaussian kernel function: $k(y, y') \propto e^{\frac{(y-y')^2}{2\sigma^2}}$. We use MSE loss, so the re-weighted loss function is given by Eq. (2), where $\hat{y}$ is the predicted value of the network and $y'$ is the value of the label.

$$loss(\hat{y}, y') = \frac{1}{n}\sum_{i=1}^{n}\frac{1}{\sqrt{\tilde{p}(y'_i)}}(\hat{y_i} - y'_i)^2 \tag{2}$$

## 4 Experiments

### 4.1 Tasks and Datasets

We evaluate our model on public **CircuitNet** [5] and **ISPD2015** [4] datasets. For the CircuitNet dataset, we focus on the two cross-stage prediction tasks, **congestion prediction** and **design rule check (DRC) violation prediction**. For the ISPD2015 dataset, our main focus lies in its congestion prediction task.

### 4.2 Baselines and Settings

We compare our work with two mainstream methods in this field: CNN-based methods and GNN-based methods. The CNN-based method mostly consists of hand-crafted features as input, extracting features using the backbone, and finally adding a segmentation header for pixel-wise prediction. CircuitNet dataset provides **Gpdl** [24] and **RouteNet** [38] as baselines for congestion prediction and DRC violation prediction tasks, respectively. Gpdl is a typical CNN-based EDA-customized deep learning model for congestion prediction, it combines RUDY map, pin RUDY map, and macro region as input features. Similarly, RouteNet is a network customized for DRC violation prediction. In addition to the three hand-crafted features mentioned above, it also incorporates cell density and congestion map as input. In this paper, input features of all CNN-based methods follow these settings. For the GNN-based method, we chose the most outstanding models in the field: **CircuitGNN** [40], a heterogeneous graph method where topological and geometrical information are integrated jointly. We evaluate the results in Pearson/Spearman/Kendall correlation as with [40, 14].

The network is trained end-to-end on a single NVIDIA RTX 3090 GPU, for 100 epochs with a cosine annealing decay learning rate schedule [26] and 10-epoch warmup. We use the AdamW optimizer with learning rate $\gamma = 0.001$.

### 4.3 Result of Congestion Prediction

**CircuitNet.** The results are summarized in Table 1. The result shows that our method achieves the best performance, and outperforms the baseline model Gpdl provided by CircuitNet, with an average metric improvement of $3.8\%$. Our method beats the UNet++ ($2.8\%$ on average), which means that the features extracted by our method are superior to the hand-crafted features, as our decoder is the same as UNet++. The performance of CircuitGNN on CircuitNet is not very good, possibly because CircuitGNN requires node-level labels for training, but the CircuitNet dataset only provides grid-level labels. We assigned the grid label to each node within the grid as the node label. During validation, we performed average pooling on the outputs of nodes within the same grid to obtain the grid-level output, which is consistent with the setting in CircuitGNN.

Fig. 3 visualizes the predicted congestion map generated by different methods. It can be intuitively seen that our method can provide more detailed predictions for the congestion map.

**ISPD2015.** The results are summarized in Table 2, it indicates that our model still achieves the best performance on the ISPD2015 dataset. It is worth noting that the ISPD2015 dataset used in our study

Table 1: Congestion prediction result of CircuitNet dataset

| Method | pearson | spearman | kendall |
|---|---|---|---|
| Gpdl [24] | 0.5032 | 0.5143 | 0.3787 |
| Gpdl with LinkNet [6] | 0.5078 | 0.4905 | 0.3623 |
| Gpdl with MANet [12] | 0.4856 | 0.5167 | 0.3797 |
| Gpdl with DeepLabv3 [7] | 0.5091 | 0.5211 | 0.3834 |
| Gpdl with Deeplabv3plus [8] | 0.5153 | 0.5160 | 0.3807 |
| Gpdl with UNet [32] | 0.5770 | 0.5103 | 0.3780 |
| Gpdl with UNet++ [46] | 0.6085 | 0.5202 | 0.3855 |
| CircuitGNN [40] | 0.3287 | 0.4483 | 0.3688 |
| CircuitFormer (ours) | **0.6374** | **0.5282** | **0.3935** |

is provided by the developers of the CircuitNet dataset. However, the format they provide is not easily compatible with CircuitGNN, especially when dealing with large-scale designs. Consequently, a significant amount of time is required for data conversion in such cases. Therefore, when comparing with CircuitGNN, we excluded the superblue design from ISPD2015, which contains over one million nodes. The experimental results are presented in Table 3, similarly, we also showcase the performance of CNN-based models.

Table 2: Congestion prediction result of ISPD2015 dataset

| Method | pearson | spearman | kendall |
|---|---|---|---|
| Gpdl [24] | 0.4345 | 0.1297 | 0.0978 |
| Gpdl with UNet [32] | 0.4173 | 0.1430 | 0.1084 |
| Gpdl with UNet++ [46] | 0.3877 | 0.1575 | 0.1189 |
| CircuitFormer (ours) | **0.4456** | **0.1797** | **0.1360** |

Table 3: Congestion prediction result of ISPD2015 dataset without superblue

| Method | pearson | spearman | kendall |
|---|---|---|---|
| Gpdl [24] | 0.5202 | 0.1880 | 0.1423 |
| Gpdl with UNet [32] | 0.5253 | 0.1481 | 0.1124 |
| Gpdl with UNet++ [46] | 0.4593 | 0.1862 | 0.1423 |
| CircuitGNN [40] | 0.3940 | 0.1912 | 0.1614 |
| CircuitFormer (ours) | **0.6534** | **0.2244** | **0.1710** |

## 4.4 Result of DRC violation prediction

Table 4 shows that in the DRC violation prediction task, our work achieved the best performance in terms of Spearman and Kendall, but the Pearson correlation coefficient is relatively low. We speculate that this is because hand-crafted methods use the congestion map as an input feature, which contains higher-level information than the raw data. In contrast, our method only uses the original positional information as input to maintain consistency.

## 4.5 Ablation study

We conducted a series of ablation experiments to demonstrate the roles of different components in our model.

**Comparison with other point cloud perception methods.** Table 5 shows a comparison between our method and other point cloud perception methods, where we replace our encoder with them. We compared point-based methods, such as **Point-Transformer** [43] and **Msg-Transformer** [13], and voxel-based (or grid-based for 2D circuit) methods, such as **PointPillars** [22] and **DSVT** [37]. The results show that the voxel-based method outperforms the point-based method in both tasks, which

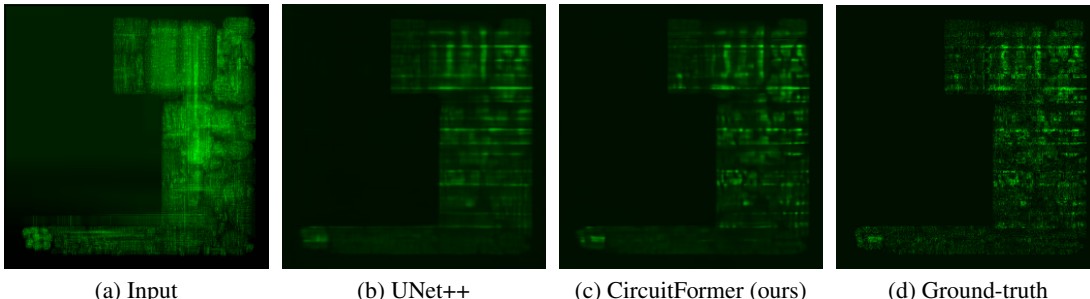

| (a) Input | (b) UNet++ | (c) CircuitFormer (ours) | (d) Ground-truth |

Figure 3: Visualization of congestion maps for CircuitNet dataset

Table 4: DRC violation prediction result of CircuitNet dataset

| Method | pearson | spearman | kendall |
|---|---|---|---|
| RouteNet [38] | 0.4206 | 0.2462 | 0.1990 |
| RouteNet with LinkNet [6] | 0.4165 | 0.2213 | 0.1788 |
| RouteNet with MANet [12] | 0.3420 | 0.2395 | 0.1916 |
| RouteNet with DeepLabv3 [7] | 0.3760 | 0.2480 | 0.1989 |
| RouteNet with Deeplabv3plus [8] | 0.4224 | 0.2699 | 0.2185 |
| RouteNet with UNet [32] | 0.4415 | 0.2746 | 0.2225 |
| RouteNet with UNet++ [46] | **0.4698** | 0.2782 | 0.2261 |
| CircuitGNN [40] | 0.1171 | 0.2366 | 0.2095 |
| CircuitFormer (ours) | 0.3707 | **0.3094** | **0.2471** |

may be due to the fact that the voxel-based method directly partitions the circuit design into grids, which is consistent with the labeling modality, whereas the point-based method needs to project the point-wise information to the grid-wise prediction, which may result in information loss. Additionally, the point-based method requires point sampling and cannot utilize information from all points. Our method also outperforms PointPillars, which groups points within the same grid and extracts features using MLP and Maxpooling like PointNet [30]. Similar to RUDY, this method can only extract local features within the grid and cannot exchange features across grids. It highlights the importance of information exchange across grids.

Table 5: Comparison with other point cloud perception methods

| Method | congestion | | | DRC violation | | |
|---|---|---|---|---|---|---|
| | pearson | spearman | kendall | pearson | spearman | kendall |
| Msg-Transformer | 0.3634 | 0.4033 | 0.2930 | 0.1216 | 0.2155 | 0.1686 |
| Point-Transformer | 0.4086 | 0.4599 | 0.2930 | 0.1307 | 0.2186 | 0.1721 |
| PointPillar | 0.6151 | 0.5234 | 0.3887 | 0.3617 | 0.3067 | 0.2447 |
| DSVT | 0.6035 | 0.5121 | 0.3802 | 0.2380 | **0.3213** | **0.2557** |
| CircuitFormer (ours) | **0.6374** | **0.5282** | **0.3935** | **0.3707** | 0.3094 | 0.2471 |

**Label distribution smooth.** Table 6 investigated the improvement brought by the label distribution smooth module on imbalanced prediction tasks.

**Preprocessing time.** In Fig. 4 we show the relationship between the preprocessing time of various methods and the size of the circuit design. GNN-based methods, represented by CircuitGNN, require generating a computation graph based on the geometric and topological information of the circuit design during the preprocessing stage. The time of this process is proportional to the size of the circuit design. CNN-based methods require hand-crafted methods to extract geometric information, which involves traversing the position of each node and pin. This process is proportional to the number of

Table 6: Results on the influence of label distribution smoothing

| Method | Prediction Task | pearson | spearman | kendall |
|---|---|---|---|---|
| CircuitFormer (w/o. LDS) | congestion | 0.6360 | 0.5143 | 0.3833 |
| CircuitFormer | congestion | **0.6374** | **0.5282** | **0.3935** |
| CircuitFormer (w/o. LDS) | DRC violation | 0.3552 | 0.2944 | 0.2354 |
| CircuitFormer | DRC violation | **0.3707** | **0.3094** | **0.2471** |

nodes and pins. In contrast, our method requires no preprocessing and takes the raw position of each node as input directly. The time tests were conducted on AMD EPYC 7502P 32-Core 2.5GHz CPU.

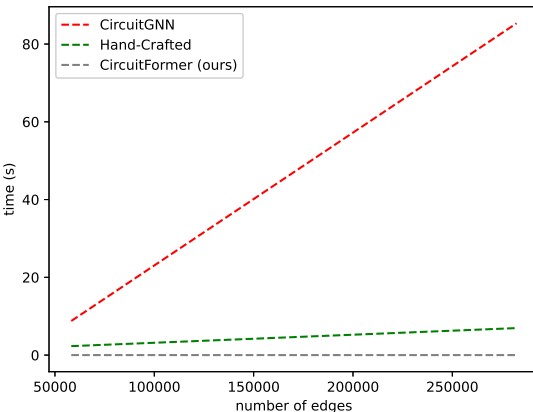

Figure 4: The relationship between preprocessing time and edge number.

**Balancing model performance and runtime memory.** To delve deeper into the relationship between the performance of our model and its runtime memory, and to provide more options to the community, we conducted ablation experiments on the encoder of our model. As shown in Fig. 1, our encoder consists of four stages of GA modules. We tested the performance and runtime memory of models with different numbers of stages. Specifically, we test runtime memory on RISCY-FPU-a, which is the largest-scale design in the CircuitNet dataset and consists of 77,707 nodes. The results are summarized in Table 7.

Table 7: The performance of models with different numbers of stages on the CircuitNet congestion prediction task, and their runtime memory during the processing of RISCY-FPU-a, which consists of 77,707 nodes.

| Method | pearson | spearman | kendall | Runtime Memory/MB |
|---|---|---|---|---|
| CircuitFormer (4-stage) | **0.6374** | **0.5282** | **0.3935** | 2475 |
| CircuitFormer (3-stage) | 0.5793 | 0.4977 | 0.3683 | 1519 |
| CircuitFormer (2-stage) | 0.5322 | 0.4972 | 0.3668 | 1047 |
| CircuitFormer (1-stage) | 0.5260 | 0.4853 | 0.3582 | **811** |
| CircuitGNN [40] | 0.3287 | 0.4483 | 0.3688 | 1501 |
| Gpdl [24] | 0.5032 | 0.5143 | 0.3787 | 1887 |
| Gpdl with UNet++ [46] | 0.6085 | 0.5202 | 0.3855 | 2209 |

**Decoder.** We conducted ablation experiments on the decoder selection to explore its impact on the model's performance.

Table 8: The ablation experiments on the decoder of congestion prediction task

| Decoder | pearson | spearman | kendall | Lantency/ms |
|---|---|---|---|---|
| UNet++ | 0.3574 | 0.4622 | 0.3407 | 26.21 |
| ResNet-18 and UNet++ | 0.6374 | **0.5282** | **0.3935** | 32.02 |
| ResNet-34 and UNet++ | 0.6082 | 0.5260 | 0.3911 | 34.71 |
| ResNet-50 and UNet++ | **0.6406** | 0.5243 | 0.3887 | 36.64 |
| ResNet-18 and UNet | 0.5237 | 0.5050 | 0.3731 | 29.90 |
| ResNet-18 and DeepLabv3plus | 0.5278 | 0.5222 | 0.3878 | 28.43 |
| ResNet-18 and LinkNet | 0.5080 | 0.4816 | 0.3553 | 29.34 |

## 5   Conclusion and Future Work

We present a new perspective on circuit design, treating circuit components as point clouds. Based on this, we propose an improved multi-scale grid-based attention module based on VoxelSet [16], which achieves SOTA results on CircuitNet and ISPD2015 datasets. Furthermore, our work can serve as a general-purpose feature extractor for chip design and can be adapted to a wider range of downstream tasks. This is the first work to transform circuit design evaluation into a point cloud perception task, presenting a novel approach to addressing the issue. It has established a bridge between the relatively mature point cloud perception methods and the still developing EDA methods, enabling us to apply methods and conclusions that have been proven effective in point cloud perception tasks to EDA tasks.

However, it should be noted that although our approach can model the relationship between points through the attention mechanism, it still does not use the prior knowledge of topology directly. Graphormer [41] encodes the topological relationships between nodes into the attention matrix, which enables the fusion of topological and geometric information in the transformer. How to incorporate the prior topological information into our model will be our future research direction.

Last but not least, the methods we use are simple and basic ones, while the GNN-based or CNN-based methods we compare against have been iterated in EDA for several years. However, thanks to the active Transformer community, there are many more advanced methods available. For example, LongNet [10] and RetNet [34] not only reduce the computational complexity to linear but also improve accuracy. With the rise of Large Language Model (LLM), the use of Transformers for handling long sequential data (corresponding to large-scale circuit designs in the EDA field) has become a trend. There has been a wealth of research dedicated to improving the performance of Transformers while reducing computational complexity. We hope to bring this trend into the EDA domain through our work, leveraging the advancements in Transformers to enhance circuit design tasks.

**Acknowledgement**: This project was supported by National Natural Science Foundation of China (NSFC No. 62276108).

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
