# OpenReview forum: "Circuit as Set of Points"
_NeurIPS.cc/2023/Conference — NeurIPS 2023 poster_

### Official Review · Reviewer_SgH5 · 2023-06-18

**Soundness:** 3 good
**Presentation:** 2 fair
**Contribution:** 3 good
**Rating:** 7
**Confidence:** 3

**Summary:**

This paper introduces a novel perspective on circuit design by treating circuit components as point clouds and utilizing point cloud perception methods for feature extraction. Previous approaches either relied on hand-crafted methods to transform circuit designs into images for CNN feature extraction or employed GNNs for graph-based feature extraction, which required time-consuming preprocessing. In contrast, the proposed method enables direct feature extraction from raw data without preprocessing, allowing for end-to-end training and achieving high performance. Experimental results on the CircuitNet dataset demonstrate the superiority of the proposed method in congestion prediction and design rule check (DRC) violation prediction tasks. This approach establishes a connection between point cloud perception methods and EDA algorithms, facilitating the application of collective intelligence in solving circuit design tasks. The method outperforms hand-crafted features and CNN-based methods in feature extraction while eliminating the need for preprocessing, thereby reducing processing time. The work shows potential as a universal feature extractor for chip design, adaptable to a wide range of downstream tasks.

**Strengths:**

The paper introduces a groundbreaking method that addresses circuit congestion prediction and design rule check (DRC) violation prediction in electronic design automation (EDA). By leveraging the concept of point clouds and incorporating a genetic algorithm (GA) module, the proposed approach achieves significant improvements over existing techniques commonly employed in EDA tools. The utilization of point cloud perception enables a fresh perspective on circuit design analysis, allowing for more accurate and efficient prediction of congestion and violations. Additionally, the integration of the GA module enhances the optimization process, leading to enhanced results and performance. This novel method showcases the potential for innovative approaches in EDA, opening doors to advanced techniques that can better handle circuit design challenges and contribute to improved circuit performance and reliability.


**Weaknesses:**

To enhance the comprehensiveness of the paper, it would be valuable to expand the evaluation to encompass other applications within the domain of electronic design automation (EDA), such as circuit placement. By investigating the effectiveness of the proposed method in various EDA tasks, including placement, the authors can provide a more holistic understanding of its capabilities and potential impact.
Furthermore, conducting additional ablation studies that delve into the runtime and scaling model trade-offs would contribute to a deeper analysis of the proposed approach. These studies would shed light on the performance and efficiency aspects of the method, allowing for a better understanding of its behavior under different conditions and dataset sizes. This information would be particularly useful for practitioners and researchers aiming to apply the method in real-world scenarios.
Another aspect worth exploring is the authors' emphasis on the encoder while leaving the decoder section with limited analysis. This raises intriguing questions regarding the possibility of modifying the output format to potentially improve the EDA software pipeline. Investigating alternative output formats or exploring ways to optimize the decoder component could uncover opportunities for further enhancing the proposed method's integration into EDA workflows.
Moreover, the paper opens up avenues for considering the incorporation of this approach into other areas of EDA tools beyond congestion prediction and DRC violation prediction. Exploring how the proposed method could be leveraged in other stages of the EDA process, such as routing, power optimization, or physical design verification, would demonstrate its versatility and potential for broader application. This could foster innovation in multiple facets of EDA, leading to more efficient and advanced chip designs.
By addressing these suggestions, the paper would offer a more comprehensive evaluation, deeper insights into runtime and scaling trade-offs, considerations for optimizing the decoder section, and opportunities for applying the proposed approach in various EDA tool domains. These extensions would further solidify the paper's contribution and provide valuable guidance for future research and development in the EDA field.

**Questions:**

A few questions:

In figure 5 shows, 3x model size than unet. Which creates the question of scaling model. If a smaller version of the model with same size of Unet will get better accuracy?

In figure 4 shows preprocessing times, what is the total runtime and inference time of the model?

**Limitations:**

The authors addressed the limitations correctly.

---

> ### Author Rebuttal · Authors · 2023-08-08
>
> We would like to thank you for your detailed comments to help us improve our work, and we will improve our manuscript correspondingly. The responses to the main concerns are as follows.
>
> **Question 1**：In figure 5 shows, 3x model size than unet. Which creates the question of scaling model. If a smaller version of the model with same size of Unet will get better accuracy? Furthermore, conducting additional ablation studies that delve into the runtime and scaling model trade-offs would contribute to a deeper analysis of the proposed approach.
>
> **Response 1**: As shown in Figure 1 of our paper, our encoder consists of four stages of GA modules, each with different grid sizes to integrate multi-scale information. We test the performance and latency of models with different numbers of stages, aiming to provide the community with more options.
>
> |Number of Stages|pearson|spearman|kendall|Latency/ms|
> |-|-|-|-|-|
> |1-stage|0.5260|0.4853|0.3582|12.15|
> |2-stage|0.5322|0.4972|0.3668|15.70|
> |3-stage|0.5793|0.4977|0.3683|23.21|
> |4-stage|**0.6374**|**0.5282**|**0.3935**|32.21|
>
> Table R1: The performance and latency of models with different numbers of stages
>
> **Question 2**：In figure 4 shows preprocessing times, what is the total runtime and inference time of the model?
>
> **Response 2**: Here we present the preprocessing time and inference time of our method compared to CNN-based method (Gpdl[22]) and GNN-based(CircuitGNN[39]) method at different scales of component numbers.
>
> |Method|Number of Nodes|Preprocessing Time/s|Inference Time/s|Total Time/s|
> |-|-|-|-|-|
> |Gpdl|29521|0.204|0.005|0.209|
> |CircuitGNN|29521|40.43|0.008|40.438|
> |Ours|29521|0|0.032|0.032|
> |Gpdl|149655|0.906|0.005|0.911|
> |CircuitGNN|149655|756.2|0.030|756.23|
> |Ours|149655|0|0.096|0.096|
>
> Table R2:  The time cost of various methods on circuit designs of different scales
>
> The experiment shows that our model achieves a shorter overall time compared to others, accomplishing the goal of reducing design cycles.
>
> The time tests were conducted on AMD EPYC 7502P 32-Core 2.5GHz and NVIDIA RTX 3090.
>
> **Question 3**: To enhance the comprehensiveness of the paper, it would be valuable to expand the evaluation to encompass other applications within the domain of electronic design automation (EDA), such as circuit placement. By investigating the effectiveness of the proposed method in various EDA tasks, including placement.
>
> **Response 3**: Thank you for your suggestion, which is very important to enhance the impact of our work in the EDA field. While the experimenting with circuit placement may use reinforcement learning methods, which is beyond the scope of our paper, we'd be happy to discuss the possibility with you. To discuss this issue, let's first have a brief understanding of how the circuit placement task is addressed. In simple terms, circuit placement takes the position of the circuit as input and minimizes the minimum wire length and layout density. This can be represented by Eq.(1)，where $\min \left(\sum WL(x,y)\right)$ is the minimum wirelength, and$D\left(x,y\right)$ is the density penalty. We can add congestion rate as a penalty term to the optimization function, which$L\left(x,y\right)$stands for congestion penalty.
> $$\min\left(\sum WL(x,y)\right)+\lambda D\left(x,y\right)+\gamma L\left(x,y\right)\tag{1}$$
> The effectiveness of this approach is demonstrated in Gpdl, where the placement method with the addition of congestion penalty achieved up to a 9.05% reduction in congestion rate and a 5.30% reduction in wire length compared to the SOTA. Compared to Gpdl, our model offers more accurate congestion predictions, with the potential to further reduce congestion and wire length in the layout.
>
> **Question 4**：Another aspect worth exploring is the authors' emphasis on the encoder while leaving the decoder section with limited analysis. Investigating alternative output formats or exploring ways to optimize the decoder component could uncover opportunities for further enhancing the proposed method's integration into EDA workflows.
>
> **Response 4**: This question is similar to **Question 1** raised by Reviewer DbXo. Due to the limitations in the length of the response, please refer to **Response 1** we gave to Reviewer DbXo .
>
> **Question 5**：Exploring how the proposed method could be leveraged in other stages of the EDA process, such as routing, power optimization, or physical design verification, would demonstrate its versatility and potential for broader application
>
> **Response 5**: We would be delighted to discuss the role of our method in other stages of the EDA process. Our model can be effectively employed as a plug-in within corresponding algorithms to enhance their performance.
>
> **Routing and Power Optimization**: Similar to our discussion in **Response 3**, regarding routing, our model can also improve the performance of the routing algorithm by predicting wiring congestion as a penalty term for the optimization objective. For power optimization, by predicting the power of the circuit, we can directly incorporate it as an optimization term in power optimization algorithms.  For specific details, please refer to **Response 3**.
>
> **Physical Design Verification**: Physical design verification consists of three main parts: design rule check, layout versus schematic, and electrical reliability check. In our paper, we extensively discussed the performance of our model in the DRC task(please refer to Section 4.4 of the paper), where it achieved outstanding results. Based on this success, we believe that our method can deliver equally impressive performance in the other two tasks as well
>
> We hope the explanations and results above solve your concerns. Thanks for your valuable comments and looking forward to your reply.
>
> *Due to the limitations in the length of the response, we utilize the reference number from our paper directly, without including an additional citation in the rebuttal. Please refer to Section **References** of the paper.

---

> > ### Comment · Reviewer_SgH5 · 2023-08-16
> > **thanks for clarification**
> >
> > thanks for clarification and for acknowledging the points.

---

> > > ### Author Response · Authors · 2023-08-17
> > > **Thanks！**
> > >
> > > Thank you for your recognition of our work and your bonus points！

---

### Official Review · Reviewer_DbXo · 2023-07-04

**Soundness:** 3 good
**Presentation:** 3 good
**Contribution:** 2 fair
**Rating:** 6
**Confidence:** 2

**Summary:**

This paper proposes to use point cloud perception methods to extract features from the circuit by treating the circuit components as point clouds. The proposed method solves the infeasibility of end-to-end training in the CNN-based method and the limitation of time-consuming preprocessing in the GNN-based method. The evaluation result shows the effectiveness of the proposed method on CircuitNet dataset: it outperforms the mainstream CNN-based and GNN-based methods on congestion prediction and design rule check violation prediction.

**Strengths:**

1. This paper facilitates Electronic Design Automation (EDA) by using the well-researched point cloud perception methods, providing a new perspective for circuit design.
2. The encoder part, especially the multi-scale grid-based attention architecture, reduces the computational complexity, and compresses the information as well.

**Weaknesses:**

  1. Please give the runtime latency comparison with the CNN-based method to prove the proposed method reduces design cycles.

**Questions:**

1. In the decoder part, motivations for using ResNet18 and UNet++ are missing. Is the selection of network architecture in the decoder part based on the downstream tasks? For different tasks, is necessary to use different networks?

**Limitations:**

The authors addressed limitations.

---

> ### Author Rebuttal · Authors · 2023-08-08
>
> Thanks for the valuable comments and suggestions which help us improve this work. Following are our detailed responses to your concerns.
>
> **Question 1**：In the decoder part, motivations for using ResNet18 and UNet++ are missing. Is the selection of network architecture in the decoder part based on the downstream tasks? For different tasks, is necessary to use different networks?
>
> **Response 1**:Your viewpoint is correct. The choice of decoder architecture is closely linked to downstream tasks. The task provided in CircuintNet, the dataset used in this paper, is concerned with prediction for grid-level labels. For example, in the congestion prediction task, we aim to predict the congestion rate for each pixel. This task bears a resemblance to the classical task of image segmentation in computer vision. Therefore, in the decoder part, we opted for the classic combination of a backbone network with a segmentation head. This combination has been proven to be effective for pixel-wise prediction tasks.
>
> It is worth noting that our model has the potential to be applied to a wider range of downstream tasks. This is because we extract node-level features rather than pixel-level features. By replacing the decoder with other point cloud decoders, such as PointNet and Point Transformer, we can extend our model to node-level prediction tasks. This flexibility is not achievable with CNN-based models, as their inputs and outputs are grid-level (or pixel-level).
>
> The core contribution of our work is to provide a novel perspective on circuit design. The main innovation lies in the encoder, so we did not focus extensively on the design of the decoder. However, prompted by your suggestion, we realized the significance of conducting more ablation experiments on the decoder. We adopted various decoder architectures, and performed ablation experiments on the congestion prediction task. The experimental results are summarized in Table R1
>
> | Decoder                        | pearson    | spearman   | kendall    | Latency/ms |
> | ------------------------------ | ---------- | ---------- | ---------- | ----------- |
> | UNet++ [45]                    | 0.3574     | 0.4622     | 0.3407     | 26          |
> | ResNet-18 [14] and UNet++ [45]  | 0.6374     | **0.5282** | **0.3935** | 32          |
> | ResNet-34 and UNet++ [45]      | 0.6082     | 0.5260     | 0.3911     | 34.71       |
> | ResNet-50 and UNet++ [45]      | **0.6406** | 0.5243     | 0.3887     | 36.64       |
> | ResNet-18 and UNet [31]         | 0.5237     | 0.5050     | 0.3731     | 29.9        |
> | ResNet-18 and DeepLabv3plus [7] | 0.5278     | 0.5222     | 0.3878     | 28.43       |
> | ResNet-18 and LinkNet [6]       | 0.5080     | 0.4816     | 0.3553     | 29.34       |
>
> Table R1: The ablation experiments on the decoder of congestion prediction
>
> As shown in Table R1, we can observe that refining the features with ResNet before performing segmentation is necessary. This process allows the node-wise features extracted by the encoder to be adapted to the grid-wise features, which serve as the output of the decoder. Considering the trade-off between model performance and latency, we believe that ResNet-18 and UNet++ is the best choice.
>
> **Question 2**：Please give the runtime latency comparison with the CNN-based method to prove the proposed method reduces design cycles.
>
> **Response 2**: Here we present the preprocessing time and inference time of our method compared to the CNN-based method at different scales of component numbers.
>
> | Method   | Number of Nodes | Preprocessing Time/ms | Inference Time/ms | Total Time/ms |
> | -------- | --------------- | ------------------ | -------------- | ---------- |
> | Gpdl [22] | 29521           | 204              | 5.4        | 209.4    |
> | Ours     | 29521           | 0                  | 32.2        | 32.2     |
> | Gpdl [22]    | 149655          | 906.1            | 5.4         | 911.5    |
> | Ours     | 149655          | 0                  | 96.4         | 96.4     |
>
> Table R2: The time consumption of various methods on circuit designs of different scales.
>
> The experiment shows that despite our model not outperforming CNN-based methods in terms of inference time, it achieves a shorter overall time compared to them, thereby accomplishing the goal of reducing design cycles. This is because CNN-based methods require extracting hand-crafted features (such as RUDY, Pin RUDY) from the circuit, necessitating time-consuming preprocessing that involves traversing the positions of all components and calculating density. In contrast, our method directly utilizes the raw positional information of the components as input, enabling end-to-end processing. The time tests were conducted on AMD EPYC 7502P 32-Core 2.5GHz CPU and NVIDIA RTX 3090 GPU.
>
> If the response satisfies your concerns, please consider reassessing the score. Thanks for your valuable comments and looking forward to your reply.
>
> *Due to the limitations in the length of the response, we utilize the reference number from our paper directly, without including an additional citation in the rebuttal. Please refer to Section **References** of the paper.

---

> > ### Comment · Reviewer_DbXo · 2023-08-20
> >
> > Thank you for your reply! Your comments addressed my concerns.

---

### Official Review · Reviewer_rfmi · 2023-07-06

**Soundness:** 3 good
**Presentation:** 3 good
**Contribution:** 3 good
**Rating:** 6
**Confidence:** 4

**Summary:**

This paper proposes a new perspective on circuit design by viewing circuit components as point clouds and using point cloud perception methods to extract features from circuits. This achieves the goals of directly extracting features from raw data, end-to-end training, and high performance. Experimental results show that the method achieves state-of-the-art performance on congestion prediction and design rule checking tasks.

**Strengths:**

a. Overall summary:
The authors propose a novel approach for chip design and manufacturing processes using Electronic Design Automation (EDA) algorithms and tools.

b. Methods:
The authors treat circuit components as point clouds and use point cloud perception methods for feature extraction.
They utilize a grid-based attention (GA) module to aggregate multi-scale and global geometric information.
The encoder converts point-wise features to grid-wise features and generates grid-wise features using a scatter-sum operator.
The decoder refines and downsamples the features extracted by the encoder using a ResNet-18 backbone and a UNet++ segmentation head.
The loss function incorporates label distribution smoothing to address imbalanced classification problems.

c.  Experimental results:
The authors conducted experiments on the CircuitNet dataset.
The proposed method achieved state-of-the-art performance in congestion prediction and design rule check (DRC) violation prediction tasks.
The method outperformed UNet in terms of performance, with a runtime memory of 3721MB for the proposed method and 1931MB for UNet.

**Weaknesses:**

N/A

**Questions:**

N/A

**Limitations:**

The limitations is not explicitly discussed, but the future work has plotted the current limitation and proposed the next direction that authors will try to tackle such limitation.
No negative societal impact

---

> ### Author Rebuttal · Authors · 2023-08-08
>
> Dear Reviewer rfmi
>
> Thank you for your recognition of our work. If you have any new questions or concerns after reviewing our discussions with other reviewers, we welcome further discussion and are available to address any additional inquiries you may have. Please feel free to reach out to us.
>
> Best wishes
>
> Authors

---

### Official Review · Reviewer_YScf · 2023-07-07

**Soundness:** 3 good
**Presentation:** 3 good
**Contribution:** 2 fair
**Rating:** 4
**Confidence:** 4

**Summary:**

The manuscript presents several claims. Firstly, it introduces a novel perspective on "point cloud perception methods" for performing EDA tasks. By treating circuit components as point clouds and utilizing point cloud perception networks, the paper proposes an innovative approach to solving these tasks. Secondly, it claims superiority over CNN and GNN-based methods commonly used for EDA tasks. The authors assert that their method outperforms these existing techniques. Thirdly, the paper positions itself as the state-of-the-art solution for EDA tasks. The experimental section is well-written, easy to understand, and the overall presentation flow of the manuscript is good.

**Strengths:**

In terms of originality, paper uses novel perspective on the solving the EDA problems which novelty in the paper. clear description of the method, and well-structured experimental section, which includes an ablation study to enhance the validity

**Weaknesses:**

The experimental results provided in the paper are not sufficient. For instance, Table1 shows the comparison of congestion with CNN and GNN methods. However, in case of CNN the cited paper uses ISPD2015 dataset (Open source in circuitnet), while this paper uses circuitnet. It may be better first show comparison with Circuitnet paper
https://www.sciengine.com/SCIS/doi/10.1007/s11432-022-3571-8
Similarly, CircuitGNN[39] which uses the ISPD2011 dataset while this paper uses Circuitnet dataset. Maybe add another comparison point cloud perception methods vs CircuitGNN on ISPD 2011 dataset.


**Questions:**

 1. Add more comparison SOTA GNN & CNN based method comparisons for both congestion and DRC violation prediction

2. Show results on other open-source datasets like (ISPD2015 which is available in circuitnet with grid level labels) and DAC 2012.

3. The paper mentions “performance is higher without excessive GPU memory”  but the memory required here is approximately double that Unet. If authors could show the comparison of memories size of graph and memory required by your method and SOTA methods would be fair.

4. The paper mentions that “novel perspective on circuit design” and “work has the potential to serve as a universal feature extractor for chip design” But only showed two tasks. To make these claim needs to perform other tasks in circuit design like power prediction, and delay prediction.


**Limitations:**

•	The scope of idea is may be limited by the size of the graph. For example, if the graph contains millions of nodes and edges which requires large GPU memory to process.
•	Generalization of the method presented in paper is explored it might be the limitation since it only works on grid-based dataset. But GNN is more generalized to solve EDA problems since netlist is a graph.

---

> ### Author Rebuttal · Authors · 2023-08-08
>
> We would like to thank you for your detailed comments to help us improve our work, and we will improve our manuscript correspondingly. The responses to the main concerns are as follows.
>
> Firstly, I would like to apologize for the inconvenience caused. Due to the word limit imposed on rebuttal, to provide a more comprehensive response to your questions, we have included the results tables in global rebuttal. Please evaluate our response in conjunction with the information provided in the global rebuttal, thanks!
>
> **Question 1**: Show results on other open-source datasets like ISPD2015 and DAC 2012.
>
> **Response 1**: Firstly, I would like to explain why we did not report results on the ISPD or DAC datasets. Both the ISPD and DAC datasets only provide raw topological information, requiring us to generate the placement ourselves. This step introduces randomness, which can bring challenges for fair comparisons(for more details, please refer to Section 4.1 of our paper).
>
> Fortunately, as you mentioned, in an update, the CircuitNet dataset developer provided the ISPD2015 dataset with complete layout information and congestion labels generated by them. To respond to your concerns regarding the generalization of our model, we conducted experiments on the ISPD2015 congestion prediction task. The results are shown in Table G1 in global rebuttal.
>
> **Question 2:** Add more comparison SOTA GNN & CNN based method comparisons for both congestion and DRC violation prediction.
>
> **Response 2:** Here, we have conducted additional comparative experiments on the CircuitNet dataset. The results are shown in Table G2-1 and Table G2-2 in global rebuttal.
>
> **Question 3:** The paper mentions “performance is higher without excessive GPU memory” but the memory required here is approximately double that Unet. If authors could show the comparison of memories size of graph and memory required by your method and SOTA methods would be fair.
>
> **Response 3:** Thanks for your kind suggestion. We want to point out that this statement means our model does not require too large GPU memory and can be inferred on a single low-end GPU (like GTX 1080ti). We will clarify our descriptions in the revision to avoid misunderstanding. We have already compared our runtime memory with other SOTA models, please refer to Figure 5 of our paper for more details. We have conducted a study on the relationship between the runtime memory of our model and circuit scale. The results are shown in Table G3 in global rebuttal. The results indicate that the runtime memory of our model exhibits a linear increase with the circuit size, which aligns with the computed complexity of $O(nkd)$ as derived in Section 3.1 of our paper.
>
> **Question 4:** The paper mentions that “novel perspective on circuit design” and “work has the potential to serve as a universal feature extractor for chip design” But only showed two tasks. To make these claim needs to perform other tasks in circuit design like power prediction, and delay prediction.
>
> **Response 4:** Thank you for your reminder. We have performed additional tasks on our model to broaden its impact. For **power prediction**, CircuitNet offers a similar task known as IR drop. IR drop is defined as the deviation of voltage from reference (VDD, VSS). We compare our model with MAVIREC, which serves as the baseline in the CircuitNet dataset. The experimental results are shown in Table G4 in the global rebuttal.
>
> It is worth noting that although our model's performance in the IR drop task is slightly inferior to the baseline, MAVIREC uses five power maps as its input, which are generated with power report from Innovus(an EDA tool). These features contain a much higher level of information than the raw data. In contrast, our model only uses the original position as input. We have reasons to believe that if we can have node-wise power as input, our model can achieve better performance.
>
> **Delay prediction** is to predict the delay of each net (the connections between nodes), while our model does not explicitly model it. However, our model is based on Transformer, where the attention map implicitly captures the relationships between nodes. Our model holds potential for accomplishing this task, and we will take it as our future work, to expand our work to a wider range of EDA domains.
>
> **Question 5:** The scope of idea is may be limited by the size of the graph. For example, if the graph contains millions of nodes and edges which requires large GPU memory to process.
>
> **Response 5:** Thanks for your concern, which has provided us with an intriguing idea to delve deeper into the impact of the scaling model on performance and runtime memory. As shown in Figure 1 of our paper, our encoder consists of four stages of GA modules. We tested the performance and runtime memory of models with different numbers of stages. Specifically, we test runtime memory on superblue12, which is the largest-scale design in the dataset and consists of 1,287,038 nodes. The results are shown in Table G5 in global rebuttal.
>
> Users have the flexibility to choose models of different scales according to their requirements.
>
> **Question 6:** Generalization of the method presented in paper is explored it might be the limitation since it only works on grid-based dataset.
>
> **Response 6:** We would like to provide some clarification on our work. Our model is not limited to grid-based datasets. The encoder in our model extracts node-wise features rather than pixel-wise features. This means that both the input and output shapes are the same, represented as (N, D), where N is the number of nodes. We can modify our decoder based on the requirements of downstream tasks. If we need to do node-level prediction, we can simply replace the decoder with point cloud decoders such as PointNet[29].
>
> If the clarifications above satisfy your concerns, please consider reassessing this paper. Thanks for your valuable comments and we are expecting further discussions with you.

---

> > ### Comment · Reviewer_YScf · 2023-08-18
> >
> > Thank you for your responses. I still have some concerns.
> > 1. > work has the potential to serve as a universal feature extractor for chip design
> >
> > Table G3, all results are worse than MAVIRE in terms of pearson, spearman, kenall. Please justify the claim of "universal feature extractor".
> >
> > 2. The CircuitGNN results in Table G1 seems to be worse than its original paper on ISPD 2015 dataset. Table G1 also missed other important references (e.g., HybribNet [a]).
> > In Table G1, the pearson, spearman, Kendall for CircuitGNN are 0.394, 0.1912, 0.1614 while in work [a], the pearson, spearman, Kendall for CirctuiGNN (they used "NetlistGNN" to refer the same paper) are 0.413, 0.216, 0.189 (all are higher than yours). In addition, work [b] has a better performance (0.271 on Spearman, 0.220 on Kendall) than yours (0.2244 on Spearman, 0.1710 on Kendall).
> >
> > [a] HybridNet: Dual-Branch Fusion of Geometrical and Topological Views for VLSI Congestion Prediction,
> > https://arxiv.org/abs/2305.05374
> >
> > 3. >Table G5: The performance of models with different numbers of stages on the CircuitNet congestion prediction task, and their runtime memory during the processing of superbule12, which consists of 1,287,038 nodes.
> >
> > It is not correct. Superblue12 is NOT in the CircuitNet congestion prediction task. superblue12 is from ISPD 2015.
> >
> > 4. The scalability of this paper is much lower compared to SOTA. The nearly doubled memory ("As a quantitative reference, the runtime memory of ours is 3721MB, while that of UNet is 1931MB") overhead brought from this paper will not be able to handle large graphs. Calling "without consuming excessive GPU memory" is not appropriate.
> >
> > 5. In addition, there are many large graphs in the EDA process, such as high-bit CSA multipliers, e.g., in GAMORA [b], that requires multi GPUs to process. In those graphs, the proposed methods in this paper would not work.
> >
> > [b] Gamora: Graph Learning based Symbolic Reasoning for Large-Scale Boolean Networks, DAC, 2023
> >
> > 6. Figure 2 lacks details. It is the visulazation of what dataset?

---

> > > ### Author Response · Authors · 2023-08-19
> > > **Thanks and Response to Reviewer YScf (1/2)**
> > >
> > > Thank you for your feedback. We are glad to have addressed some of your concerns. Regarding your further questions, we would like to provide some clarifications.
> > >
> > > **Response to Question 1:** As mentioned in **Response 4** in our first reply, MAVIREC uses five power maps as its input, including internal power map, switching power map, toggle rate scaled power map, all power map and time-decomposed power map. These power maps are generated with a power report from Innovus(an EDA tool). These features contain a much higher level of information than the raw position data.  However, since the CircuitNet dataset does not provide a node-wise power report, our model only uses the original position as input. Even so, our model's performance is very close to that of MAVIREC. This also demonstrates that our method serves as a universal feature extractor, we do not rely on high-level power reports, which are not easily obtainable at a low cost. This flexibility also showcases the universality of our model. We have reasons to believe that if we can have node-wise power as input, our model can achieve better performance than MAVIREC.
> > >
> > > **Response to Question 2:** Due to the complexity of the issues raised, we will address your questions sentence by sentence for clarity and accuracy.
> > >
> > > > The CircuitGNN results in Table G1 seems to be worse than its original paper on ISPD 2015 dataset.
> > >
> > > The dataset used in the original CircuitGNN paper was ISPD2011, not ISPD2015, which we used in our study.
> > >
> > > > Table G1 also missed other important references (e.g., HybribNet [a]). HybribNet  has a better performance
> > >
> > > Firstly, HybribNet first submitted on arxiv on May.7th 2023, just ten days before the NeurIPS 2023 deadline (May.17th 2023), according to NeurIPS 2023 call for papers(https://neurips.cc/Conferences/2023/CallForPapers), it is our contemporaneous work, we should not be asked to compare our work with it.
> > >
> > > Furthermore, we can also provide an explanation for the results of HybribNet. In HybribNet, they set the number of training epochs to 500, while in our paper, all models were trained for only 100 epochs (for more details, please refer to Section 4.2 of our paper). Therefore, directly comparing our model to theirs would be unfair.
> > >
> > > > the pearson, spearman, Kendall for CirctuiGNN in HybribNet [a] are higher than CirctuiGNN in your rebuttal
> > >
> > > Similar to the previous point, in HybribNet, the CircuitGNN was trained for 500 epochs.  However, for a fair comparison, all models in our paper, including CircuitGNN, were trained for only 100 epochs, which aligns with the settings specified in the original CircuitGNN paper.
> > >
> > > **Response to Question 3:** We would like to provide an explanation for the results in Table G5. We are aware that superblue12 is a design from the ISPD2015 dataset. We chose to use it for the runtime memory test because it is the largest design among all the designs in the ISPD2015 dataset and the CircuitNet dataset. By selecting it as an extreme case, we aim to demonstrate that our model can handle such challenging designs, let alone smaller-scale designs.
> > >
> > > **Response to Question 4:** We must acknowledge that in large-scale circuit design，the runtime memory of our model is bigger than that of CNN-based models (e.g., UNet). This is because CNN-based models convert circuit designs into fixed-sized images regardless of the circuit's scale, resulting in a significant loss of fine-grained features. Additionally, they can only handle grid-level tasks.
> > >
> > > In addition, for GNN-based methods, the results presented in Table G5 in global rebuttal show that our model of both 1-stage and 2-stage versions exhibits lower runtime memory usage compared to the GNN-based SOTA(CircuitGNN) , while also delivering better performance. We believe that this evidence demonstrates that our model exhibits higher scalability compared to GNN-based SOTA

---

> > > ### Author Response · Authors · 2023-08-19
> > > **Thanks and Response to Reviewer YScf (2/2)**
> > >
> > > **Response to Question 5:** GAMORA utilizes a GNN model to handle large-scale boolean networks, as mentioned in **Response to Question 4**, our model of both 1-stage and 2-stage versions exhibits lower runtime memory usage compared to the GNN-based SOTA, so that we firmly believe that our model is capable of handling large-scale designs in GAMORA. In addition, our model is a preliminary exploration of the application of the point cloud transformer method in circuit design, using vanilla transformer. However, thanks to the active Transformer community, there are many methods that not only reduce computational complexity to linear but also improve accuracy. For example, LongNet [1] and RetNet [2], adding such methods to our model will bring better scalability to our model.
> > >
> > > With the rise of GPT, the use of Transformers for handling long sequential data (corresponding to large-scale circuit designs in the EDA field) has become a trend. There has been a wealth of research dedicated to improving the performance of Transformers while reducing computational complexity. We aim to bring this trend into the EDA domain, leveraging the advancements in Transformers to enhance circuit design tasks.
> > >
> > > [1] Ding, Jiayu, et al. "Longnet: Scaling transformers to 1,000,000,000 tokens." *arXiv preprint arXiv:2307.02486* (2023).
> > >
> > > [2] Sun, Yutao, et al. "Retentive Network: A Successor to Transformer for Large Language Models." *arXiv preprint arXiv:2307.08621* (2023).
> > >
> > > **Response to Question 6:** Thank you for your concerns. In the initial version of our paper, we only utilized the CircuitNet dataset. Therefore, Figure 3 (You refer to Figure 2, which is the architecture of grid-based attention (GA) module, we think it is a clerical error) presents the visualization of the predictions of various models on the congestion map of the CircuitNet dataset. We will clarify our descriptions in the revision to avoid misunderstanding.
> > >
> > > We hope the explanations provided above address your concerns. If they do, we kindly request you to reconsider your evaluation of this paper. Thank you for your valuable comments.

---

> > > > ### Comment · Reviewer_YScf · 2023-08-20
> > > >
> > > > Thank you for your responses.
> > > >
> > > > 1. I am aware that HybribNet [a] was a concurrent work. However, I am afraid that the comparison in this paper with a vital reference, CircuitGNN, (the most outstanding GNN model in this paper) is UNFAIR.
> > > > One of the major claims is the proposed method could achieve better accuracy than CircuitGNN. In this case, both CircuitGNN and the proposed method should report the best possible accuracy, regardless of the epochs and settings. I am also aware other settings in [a] and this work are not quite the same, e.g., [a] used "AdamW optimizer for 500 epochs with an initial learning rate 2e-4", while this work used "100 epochs with a cosine annealing decay learning rate schedule [25] and 10-epoch warmup", and "AdamW optimizer with learning rate 0.001". However, if this work claims that it achieves better accuracy than CircuitGNN, the authors should report the best possible accuracy of both CitcuitGNN and this work.
> > > >
> > > > As far as we could know, to date, the CircuitGNN results in Table G1 are worse than CircuitGNN on ISPD 2015 dataset. In Table G1, the pearson, spearman, Kendall for CircuitGNN are 0.394, 0.1912, 0.1614 (100 epochs); while in work [a], the reported results of CirctuiGNN are pearson, spearman, Kendall are 0.413, 0.216, 0.189 (all higher than this work).
> > > >
> > > > 2. >We are aware that superblue12 is a design from the ISPD2015 dataset. We chose to use it for the runtime memory test because it is the largest design among all the designs in the ISPD2015 dataset and the CircuitNet dataset.
> > > >
> > > > In the same row of Table G5, it is misleading that the performance and runtime results are from two different datasets. Please justify in the very beginning when the Table was created.
> > > >
> > > > I would keep my score.

---

> > > > > ### Author Response · Authors · 2023-08-21
> > > > > **Thanks for your feedback**
> > > > >
> > > > > Thanks for your feedback. Regarding your further questions, we would like to provide some clarifications.
> > > > >
> > > > > **Response to Question 1:**
> > > > >
> > > > > Thank you for your kind reminder. We believe that your comparative approach can provide a more comprehensive evaluation of our model. However, we also believe that our previous comparison was fair. When reproducing CircuitGNN, we followed the same settings as outlined in the original paper, with a training duration of 100 epochs. Similarly, our model was also trained for 100 epochs. Certainly, as you mentioned, it is important to compare our model with the currently reproduced SOTA CircuitGNN in HybridNet. We have already started training a version of our model for 500 epochs. However, due to the time constraints for the rebuttal, we were unable to obtain the results within the given time. We will report the results in the final version of the paper.
> > > > >
> > > > > Here, we compared the performance of our model trained for 100 epochs with CircuitGNN on the ISPD2015 dataset. The experimental results are presented in Table R1. The experimental results show that our 100-epoch-trained model outperformed the 500-epoch-trained CircuitGNN by 28.2% in terms of the average metric. We believe that with a more extensive training of 500 epochs, our model will achieve better results.
> > > > >
> > > > > | Method     | training epochs | pearson    | spearman   | kendall    | average |
> > > > > | ---------- | --------------- | ---------- | ---------- | ---------- | ------- |
> > > > > | CircuitGNN [39] | 100             | 0.3940     | 0.1912     | 0.1614     | 0.2489  |
> > > > > | CircuitGNN [39] | 500             | 0.4130     | 0.2160     | **0.1890** | 0.2727  |
> > > > > | Ours       | 100             | **0.6534** | **0.2244** | 0.1710     | **0.3496**  |
> > > > >
> > > > > Table R1. Results of models with different numbers of training epochs on the ISPD2015 congestion prediction task
> > > > >
> > > > > **Response to Question 2:**
> > > > >
> > > > > Thank you for your kind reminder. We chose superblue12 as a runtime memory test because you were concerned that our model could not handle million-level designs. However, we acknowledge that the designs in CircuitNet are relatively small, with the largest design, RISCY-FPU-a, consisting of only 77,707 nodes, could not prove that our model could handle large-scale designs. This could lead to some misunderstandings, as you say, so we report the runtime memory in processing the RISCY-FPU-a design in Table R2. The results confirm our previous conclusion that both the 1-stage and 2-stage versions of our model exhibit lower runtime memory usage compared to the SOTA GNN-based model (CircuitGNN).
> > > > >
> > > > > | Method          | pearson | spearman | kendall | Runtime Memory/MB |
> > > > > | --------------- | ------- | -------- | ------- | ----------------- |
> > > > > | Ours (4-stage)  | 0.6374  | 0.5282   | 0.3935  | 2475              |
> > > > > | Ours (3-stage)  | 0.5793  | 0.4977   | 0.3683  | 1519              |
> > > > > | Ours (2-stage)  | 0.5322  | 0.4972   | 0.3668  | 1047              |
> > > > > | Ours (1-stage)  | 0.5260  | 0.4853   | 0.3582  | 811               |
> > > > > | CircuitGNN [39] | 0.3287  | 0.4483   | 0.3688  | 1501              |
> > > > >
> > > > > Table R2. The performance of models with different numbers of stages on the CircuitNet congestion prediction task, and their runtime memory during the processing of RISCY-FPU-a, which consists of 77,707 nodes.

---

### Author Rebuttal · Authors · 2023-08-08

##

| Method         | pearson    | spearman   | kendall    |
| -------------- | ---------- | ---------- | ---------- |
| Gpdl [22]       | 0.5202     | 0.1880     | 0.1423     |
| UNet [31]       | 0.5253     | 0.1481     | 0.1124     |
| UNet++ [45]     | 0.4593     | 0.1862     | 0.1423     |
| CircuitGNN [39] | 0.394      | 0.1912     | 0.1614     |
| Ours           | **0.6534** | **0.2244** | **0.1710** |

Table G1: Congestion prediction result on ISPD2015

| Method        | pearson    | spearman   | kendall    |
| ------------- | ---------- | ---------- | ---------- |
| MANet [1*]     | 0.4856     | 0.5167     | 0.3797     |
| DeepLabv3 [2*] | 0.5091     | 0.5211     | 0.3834     |
| Ours          | **0.6374** | **0.5282** | **0.3935** |

Table G2-1:  Additional results on CircuitNet congestion prediction

| Method        | pearson    | spearman   | kendall    |
| ------------- | ---------- | ---------- | ---------- |
| MANet [1*]     | 0.3420     | 0.2395     | 0.1916     |
| DeepLabv3 [2*] | **0.3760** | 0.2480     | 0.1989     |
| Ours          | 0.3707     | **0.3094** | **0.2471** |

Table G2-2:  Additional results on CircuitNet DRC violation prediction

| Number of Nodes | Runtime Memory/MB |
| --------------- | ----------------- |
| 29521           | 1913              |
| 48127           | 2197              |
| 81643           | 2707              |
| 108292          | 3151              |
| 127419          | 3471              |

Table G3: The runtime memory of our model on circuit designs of different scales

| Method       | Input          | pearson | spearman | kendall |
| ------------ | -------------- | ------- | -------- | ------- |
| MAVIREC [3*] | Power Maps (high level)    | 0.7144  | 0.8369   | 0.7344  |
| Ours         | Nodes Position (low level) | 0.6395  | 0.829    | 0.7158  |

Table G4: IR drop prediction results

| Method         | pearson | spearman | kendall | Runtime Memory/MB |
| -------------- | ------- | -------- | ------- | ----------------- |
| Ours (4-stage) | 0.6374  | 0.5282   | 0.3935  | 22487             |
| Ours (3-stage) | 0.5793  | 0.4977   | 0.3683  | 12429             |
| Ours (2-stage) | 0.5322  | 0.4972   | 0.3668  | 7379              |
| Ours (1-stage) | 0.5260  | 0.4853   | 0.3582  | 4867              |
| CircuitGNN [39]     | 0.3287  | 0.4483   | 0.3688  | 12017             |

Table G5:  The performance of models with different numbers of stages on the CircuitNet congestion prediction task, and their runtime memory during the processing of superbule12, which consists of 1,287,038 nodes.

[1*]T. Fan, G. Wang, Y. Li and H. Wang, "MA-Net: A Multi-Scale Attention Network for Liver and Tumor Segmentation," in *IEEE Access*, vol. 8, pp. 179656-179665, 2020, doi: 10.1109/ACCESS.2020.3025372.

[2*]Chen, Liang-Chieh, et al. "Rethinking atrous convolution for semantic image segmentation." *arXiv preprint arXiv:1706.05587* (2017).

[3*]Chhabria, Vidya A., et al. "MAVIREC: ML-aided vectored IR-drop estimation and classification." *2021 Design, Automation & Test in Europe Conference & Exhibition (DATE)*. IEEE, 2021.

*Due to the limitations in the length of the response, we utilize the reference number from our paper directly, without including an additional citation in the rebuttal unless otherwise stated. Please refer to Section **References** of the paper.

---

### Decision · Program_Chairs · 2023-09-21

**Decision:**

Accept (poster)

**Comment:**

The paper proposes a novel approach to Electronic Design Automation (EDA) tasks by treating circuit components as point clouds and employing point cloud perception methods for feature extraction. The authors claim that this approach outperforms existing CNN and GNN methods in congestion prediction and Design Rule Check (DRC) violation prediction tasks.

The paper presents a novel approach to EDA tasks, which is well-received for its originality and potential impact on the field.  Considering the originality of the work, and the authors' thorough responses to the reviews, I recommend Accept for this paper. It presents a promising avenue for EDA tasks with a unique approach and has the potential for high impact in the field. However, the authors should consider extending their evaluations and providing a more comprehensive analysis to address the weaknesses identified.